# LoRDQ: activation-aware Low-Rank Decomposition and Quantization for Large Language Model Compression

## Abstract

Large language models (LLMs) deliver high performance but remain prohibitively expensive to deploy in resource-constrained environments. Post-training quantization (PTQ) is widely used to reduce memory and compute, while it often degrades sharply in the ultra-low-bit regime. Although recent PTQ methods incorporate weight sensitivity for further improvement, the sensitivity analysis is often conducted at the element-, row-, or vector-wise level within the original weight matrix, which can limit robustness at very low bitwidths. We instead operate at the *subspace* level by deriving an activation-aware low-rank factorization of each weight matrix (for a given layer/block). The key idea is to represent each weight matrix by a small set of activation-aware components that retain most output energy, and to solely quantize these factors, enabling higher precision per stored parameter under the same budget and improving accuracy in the low-bit regime. We thus propose **LoRDQ**, an activation-aware low-rank decomposition and quantization scheme that provides a closed-form factorization minimizing layer-output reconstruction, and incorporates two complementary techniques to mitigate the loss from quantizing low-rank factors, including a block-wise greedy decomposition and an intra-block compensation step. Simulations demonstrate that LoRDQ can achieve $\sim 10\times$ lower perplexity in comparison with existing methods such as GPTQ and AWQ. Moreover, leveraging our analytical results, we provide a theoretical explanation for these gains by connecting them to the spectrum of the output Gram matrix $WXX^\top W^\top$, clarifying when low-rank structure preserves critical model behavior.

## 1 Introduction

Large-scale neural networks such as large language models (LLMs) have achieved state-of-the-art performance across a wide range of applications (Roumeliotis and Tselikas, 2023). However, their massive parameter sizes and computational demands pose major challenges for deployment in resource-constrained environments, such as on-device inference or real-time applications. Post-training model compression has emerged as a key technique for alleviating these challenges, reducing both memory footprint and compute requirements without retraining.

Among various compression techniques, *post-training quantization* (PTQ) and *low-rank approximation* stand out as practical and effective solutions. PTQ reduces storage and compute by representing model parameters using low-bitwidth integers, while low-rank decomposition reduces parameter dimensionality by exploiting the inherent redundancy in weight matrices (Frantar et al., 2023; Xiao et al., 2023; Kim et al., 2023; Leconte et al., 2024). Despite significant progress in these areas, existing approaches often face limitations when used in isolation: PTQ schemes using uniform or heuristic bitwidths provide a coarse treatment of weight sensitivity, which can limit robustness when the budget is highly constrained (e.g., 2-bit). Low-rank decomposition, on the other hand, is often used in parameter-efficient fine-tuning (PEFT) or as adapters for compensating compression errors (Wang et al., 2024) , but the low-rank matrices themselves are typically left uncompressed. In fact, an efficient low-rank decomposition can extract the components with the highest energy, and these components can then be intelligently quantized, aligning with the goal of applying precision where it matters most.

To address these limitations, we propose a two-step framework for post-training model compression that combines the strengths of low-rank decomposition and quantization in a goal-oriented manner. First, we formulate the low-rank approximation problem as minimizing a task-aligned loss function based on the Frobenius norm of the output error and derive a closed-form solution using a Cholesky-based projection of the input covariance matrix. Compared to existing works (Liu et al., 2024), which requires spectral decomposition of the covariance for eigenspace projection, our approach leverages the triangular structure of the Cholesky factor, making it computationally more efficient and readily invertible. Furthermore, we prove that the derived decomposition is not only optimal for this specific projection but also achieves the same optimality for a broader class of problems where the input data matrix can be manipulated (e.g., through whitening), eliminating the need for explicit data transformations in practice.

Secondly, to compress the low-rank matrices derived from the decomposition, we propose two complementary schemes aimed at reducing performance degradation, particularly under low bitwidth constraints. The first is a *block-wise greedy decomposition* strategy, in which the target low-rank space is partitioned into multiple blocks and extracted in stages rather than in a single step. This staged procedure enables each block to be computed with explicit consideration of the quantization errors introduced in previously compressed blocks, thereby progressively refining the approximation and improving overall reconstruction fidelity. The second is an *intra-block quantization compensation* technique, which mitigates error accumulation within each block. As the basis vectors within a block are quantized sequentially, the quantization of one basis inevitably introduces residual errors that affect subsequent components. Our method compensates for these distortions by adjusting the coefficients associated with the remaining unquantized bases, thereby redistributing part of the quantization error across the block and reducing its impact on the final approximation. Together, these two schemes constitute a scalable and robust framework for compressing low-rank factors.

The main contributions of this work are threefold. First, we formulate an activation-aware low-rank decomposition that minimizes layer output error and derive a closed-form solution for any valid square-root factorization of the Gram matrix of layer inputs, proving that it achieves the same optimum as broader output-preserving formulations without requiring data matrix manipulation such as whitening. Second, we propose an integrated quantization scheme that combines a block-wise greedy decomposition with an intra-block quantization compensation strategy, enabling efficient compression of low-rank factors to mitigate quantization-induced performance degradation under aggressive bitwidth constraints. Finally, we demonstrate the strong empirical performance of our method on LLaMA-2, LLaMA-3, and OPT models, achieving superior compression–accuracy trade-offs in the ultra-low 2-bit quantization regime. We provide a spectral interpretation of these gains by connecting our approach to the spectrum of the Gram matrix of layer outputs, offering deeper insights into how the proposed decomposition preserves critical model behavior.

## 2 RELATED WORKS

### 2.1 POST-TRAINING QUANTIZATION

Post-training quantization (PTQ) converts pre-trained full-precision models into low-precision formats without requiring additional training, making it a practical and widely adopted approach for deploying large language models. In this work, we focus on *weight-only* PTQ, which avoids quantizing activations and thus simplifies deployment while maintaining low inference overhead. Representative methods such as GPTQ (Frantar et al., 2023) and SqueezeLLM (Kim et al., 2023) improve quantization accuracy using advanced rounding or blockwise optimization techniques. However, these PTQ approaches still rely on fixed or uniform bitwidth allocations, which fail to account for the heterogeneous sensitivity of weights to quantization noise, leading to notable degradation under aggressive bitwidth constraints. *Transformation-based* methods, such as QuIP (Chee et al., 2023), OmniQuant (Shao et al., 2024), and QuIP# (Tseng et al., 2024), tackle this issue by reshaping the weight or input space before quantization, significantly improving accuracy. Yet, these approaches introduce additional computational and storage overhead, particularly due to the need for large transformation matrices. An alternative is *sensitivity-based* quantization, as seen in methods such as AWQ (Lin et al., 2024), OWQ (Lee et al., 2024), SpQR (Dettmers et al., 2024), and BAQ (Zhang et al., 2025), which adaptively allocate precision based on activation statistics, Hessian-weighted losses, or layer-wise objective function minimization while adding less computational and storage

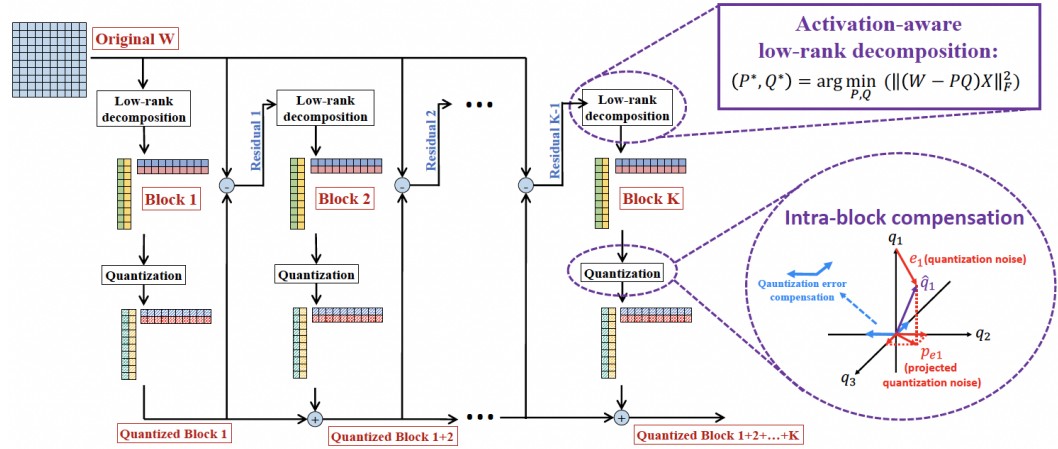

Figure 1: Overview of LoRDQ. Given a weight matrix $W$ and calibration activations $X$, we extract $W$ block-by-block by solving $\min_{P^{(k)}, Q^{(k)}} \|(R - P^{(k)}Q^{(k)})X\|_F^2$, quantizing each block with *intra-block compensation*, updating $R \leftarrow R - \widehat{P}^{(k)}\widehat{Q}^{(k)}$, and summing $\widehat{W} = \sum_{k=1}^{K} \widehat{P}^{(k)}\widehat{Q}^{(k)}$.

overhead. As an extreme compression regime, weight binarization (e.g., BiLLM(Huang et al., 2024), PB-LLM(Shang et al., 2023)) achieves sub–2-bit precision by optimizing scaling factors. In practice, the required indices for codebook/group information (often $\approx$ 1 extra bit per weight) diminish the effective compression and make deployment harder. Building on these insights, our work develops a structured low-rank factorization framework for precision-aware compression: instead of transforming the entire weight space, we identify the most informative components through activation-aware low-rank decomposition and efficiently compress them under strict bitwidth constraints.

## 2.2 Low-Rank decomposition and adaptation

Low-rank techniques are widely used for efficient adaptation: LoRA (Hu et al., 2022), QLoRA (Dettmers et al., 2023), QA-LoRA (Xu et al., 2023), and variants inject trainable low-rank adapters and *require* end-to-end finetuning, which makes them unsuitable for post-training deployment. A complementary direction uses low-rank adapters, aiming to compensate for errors introduced by quantization or pruning. Examples include EoRA (Liu et al., 2024), ASVD (Yuan et al., 2023), SVD-LLM (Wang et al., 2024), and ResSVD (Bai et al., 2025). While effective in reducing reconstruction error, these methods typically do *not* compress the low-rank factors themselves. When large ranks are used (often exceeding 20% of the original dimension), the low-rank factors are stored at full precision, resulting in nontrivial overhead that limits their efficiency under tight memory budgets. In contrast, our work considers both the low-rank decomposition and quantization, extracting the most informative components through a Cholesky-based factorization and compressing them under strict bitwidth constraints, thereby closing the gap between theoretical compression and practical efficiency.

## 3 Activation-aware Low-Rank Decomposition

Model compression for large-scale neural networks often begins with minimizing the layer-wise output discrepancy between the original and compressed weights. Specifically, given a weight matrix $W \in \mathbb{R}^{M \times N}$ (with $M$ output dimensions and $N$ input dimensions) and an input activation matrix $X \in \mathbb{R}^{N \times K}$ where $K$ denotes the number of calibration samples., the layer-wise compression objective can be formulated as:

$$\min_{\widehat{W}} \|(W - \widehat{W})X\|_F^2. \tag{1}$$

This formulation, widely adopted in post-training quantization methods such as GPTQ, directly captures the mismatch in layer outputs due to weight modifications. However, solving it exactly is

computationally prohibitive. Practical scalar-quantization approaches approximate it by sequentially quantizing entries and compensating with the temporarily unquantized weights. Recent variants further *reweight precision* using curvature or activation statistics (Hessian-aware, outlier-aware, etc.), thereby modeling heterogeneity in weight sensitivity within the original basis.

To exploit quantization resources more efficiently, a promising alternative is to model weight sensitivity at the *subspace* level, namely, approximate $W$ using a *low-rank representation*. This allows the model to retain the most informative subspace with fewer parameters, and then apply quantization to these compact components. We therefore propose a two-step framework, first find a *low-rank representation* of $W$ to preserve most significant information, and then compress these low-rank factors efficiently (will be presented in the next section).

In this section, we focus on the first step, an *activation-aware low-rank decomposition* that represents $W$ as $P \in \mathbb{R}^{M \times r}$ and $Q \in \mathbb{R}^{r \times N}$ by minimizing the layer-wise output error:

$$\min_{P \in \mathbb{R}^{M \times r}, Q \in \mathbb{R}^{r \times N}} \|(W - PQ)X\|_F^2. \tag{2}$$

This decomposition explicitly accounts for the input activation, ensuring that the retained subspace aligns with the directions most critical for maintaining layer outputs.

To enable closed-form derivation of optimal $P$ and $Q$, we assume that the matrix $XX^\top \in \mathbb{R}^{N \times N}$ is full-rank, a condition typically satisfied when the calibration or inference data exhibits sufficient variability. In practice, a small damping term $\epsilon I$ is often added to $XX^\top$ to ensure invertibility. Under this assumption, the following proposition provides the optimal solution to the activation-aware low-rank decomposition problem.

**Proposition 1.** Let $XX^\top = YY^\top$ for some invertible $Y \in \mathbb{R}^{N \times N}$. Define $W_Y := WY \in \mathbb{R}^{M \times N}$, and let the rank-$r$ truncated SVD of $W_Y$ be

$$W_Y = U_r \Sigma_r V_r^\top, \tag{3}$$

where $U_r \in \mathbb{R}^{M \times r}$, $\Sigma_r \in \mathbb{R}^{r \times r}$, and $V_r \in \mathbb{R}^{N \times r}$. Then

$$P^* = U_r \Sigma_r, \quad Q^* = V_r^\top Y^{-1} \tag{4}$$

is an optimal solution to problem equation 2. Moreover, for any valid choice of $Y$ such that $XX^\top = YY^\top$, the corresponding pair $(P^*, Q^*)$ constructed as above achieves the same minimal objective value.

In contrast with classical LoRA (Hu et al., 2022) which requires end-to-end finetuning, Proposition 1 indicates that the optimal low rank factors to minimize the output error can be derived and explicitly expressed. It also establishes that any valid square-root factorization $Y$ of $XX^\top$ leads to an optimal low-rank approximation for problem equation 2. This is because if $Y_1$ and $Y_2$ are two such factorizations, then $Y_2 = Y_1 S$ for some orthogonal matrix $S$, and the change of basis does not affect the singular values or the achieved Frobenius norm of the optimal low-rank reconstruction. Thus, the solution class is invariant under orthogonal transformations of $Y$.

This general formulation unifies several existing approaches. For instance, the EoRA (Liu et al., 2024) method corresponds to a particular choice of $Y$, taking $Y = U\Lambda^{1/2}$ from the eigendecomposition $XX^\top = U\Lambda U^\top$. While this is a valid square-root factorization, it requires a full spectral decomposition of $XX^\top$, which incurs $O(N^3)$ computational complexity and involves dense matrix operations that can be costly for large models.

In contrast, we adopt the Cholesky decomposition $XX^\top = LL^\top$ in this paper, where $L$ is lower-triangular, for constructing $Y$. This reduces the factorization cost from $O(N^3)$ for an SVD of $XX^\top$ to $O(N^3/3)$ and offers a significant computational advantage for high-dimensional activations. Additionally, the triangular structure of $L$ enables efficient inversion of $Y$ via forward and backward substitution, further improving scalability for large-scale post-training compression tasks. These properties make the Cholesky-based construction particularly attractive in practice.

While Proposition 1 provides the optimal factors $P^*$ and $Q^*$ for equation 2, a natural question arises: how does this formulation relate to the broader class of low-rank approximations that directly minimize the output error? In particular, one can consider the unconstrained problem

$$\min_{A \in \mathbb{R}^{M \times r}, B \in \mathbb{R}^{r \times K}} \|WX - AB\|_F^2, \tag{5}$$

which corresponds to the truncated SVD of $WX$. Unlike equation 2, this formulation does not impose any structure on $B$ (e.g., $B = QX$) and therefore represents the most general low-rank reconstruction of the layer output.

As highlighted in prior works such as SVD-LLM (Wang et al., 2024), the optimal solution to equation 5 can be achieved when manipulations on the activation matrix $X$ are allowed (e.g., whitening transformations). This raises a critical question for our setting: *without touching or transforming $X$, does solving equation 2 lead to a worse approximation than equation 5? If so, what is the performance gap?* Corollary 1 addresses this question, showing that our solution achieves the same optimal reconstruction of $WX$ as equation 5, thereby eliminating the need for explicit manipulation of $X$ while retaining computational efficiency.

**Corollary 1.** Let $P^*, Q^*$ be defined as in Proposition 1 using any factorization $XX^\top = YY^\top$. Then $(P^*, Q^*)$ is also an optimal solution of the broader optimization problem:

$$\min_{A \in \mathbb{R}^{M \times r}, \ B \in \mathbb{R}^{r \times K}} \|WX - AB\|_F^2, \tag{6}$$

and yields the same reconstruction as the rank-$r$ truncated SVD of $WX$:

$$P^*Q^*X = P_X Q_X, \tag{7}$$

where $P_X Q_X$ denotes the rank-$r$ truncated SVD of $WX$.

Corollary 1 shows that our solution matches the truncated SVD of $WX$, achieving the same optimal reconstruction without explicit manipulation of $X$. This eliminates the computational overhead of activation-space transformations (e.g., whitening) used in SVD-based methods, making our approach more practical for large-scale post-training compression.

## 4 COMPRESSION OF LOW-RANK FACTORS

The decomposition in Section 3 provides optimal low-rank factors $P^* \in \mathbb{R}^{M \times r}$ and $Q^* \in \mathbb{R}^{r \times N}$ that preserve the layer outputs, but these matrices are still stored in full precision by default. Without further compression, the overall parameter footprint remains dominated by $P$ and $Q$, limiting the practical savings from the decomposition.

Most existing post-training quantization (PTQ) methods directly quantize full-rank weights, often ignoring the low-dimensional subspaces that capture most of the weight energy. Conversely, SVD-based compression approaches extract these subspaces but store them in full precision, introducing non-trivial overhead that undermines their compression gains. Thus, a key challenge is *how to quantize low-rank factors effectively under low bitwidths without degrading the approximation quality*.

To address this, we propose two complementary techniques: *Block-wise greedy low-rank decomposition*, which partitions the target subspace into multiple smaller blocks and extracts them sequentially, explicitly accounting for quantization errors at each step; *Intra-block quantization compensation*, which adjusts unquantized components within a block to absorb quantization errors introduced by previously quantized components. A schematic overview of these techniques is provided in Fig. 1.

### 4.1 BLOCK-WISE GREEDY LOW-RANK APPROXIMATION

A straightforward way to obtain a rank-$r$ approximation of $W$ is to perform a single-step truncated SVD, retaining the top-$r$ singular components. While conceptually simple, this approach has a critical limitation for compression: all $r$ components are extracted jointly under the assumption of full precision, but in practice they are quantized. Once the most significant components are compressed, the induced quantization noise alters the effective residual subspace. Because single-step SVD does not account for this distortion, the subsequent components are no longer optimal for the quantized representation, leading to suboptimal reconstruction quality.

To overcome these issues, we adopt a *block-wise greedy decomposition* strategy that partitions the target rank $r$ into $K$ smaller blocks of size $r_b$ and extracts them sequentially. Unlike residual-SVD approaches designed only for truncation error correction (Bai et al., 2025), our method explicitly accounts for *quantization-induced distortion* at each stage: once a block is quantized, its contribution and associated errors are embedded into the residual, allowing subsequent blocks to adaptively

compensate. This multi-block framework effectively mitigates error accumulation and improves reconstruction fidelity under aggressive bitwidth constraints.

To facilitate a stable and activation-aware decomposition, we first incorporate the matrix $XX^\top$ into the factorization. Let $Y \in \mathbb{R}^{N \times N}$ be the Cholesky factor of $XX^\top$ such that $XX^\top = YY^\top$, and define the projected weight matrix $W_Y := WY \in \mathbb{R}^{M \times N}$. Rather than computing the full rank-$r$ truncated SVD of $W_Y$ at once, we partition the target rank $r$ into $K$ disjoint blocks of equal size $r_b$ ($r = Kr_b$) and extract them sequentially.

For block $k$, we define the residual matrix as

$$R_Y^{(k)} := W_Y - \sum_{j=1}^{k-1} \widehat{P}_Y^{(j)} \widehat{Q}_Y^{(j)}, \tag{8}$$

with initialization $R_Y^{(1)} = W_Y$, where $\widehat{P}_Y^{(j)} \in \mathbb{R}^{M \times r_b}$ and $\widehat{Q}_Y^{(j)} \in \mathbb{R}^{r_b \times N}$ are the quantized factors from previous blocks.

The detailed procedure to compute the low rank matrices $\widehat{P}^{(k)}$ and $\widehat{Q}^{(k)}$ for all $k \in \{1, \ldots, K\}$ can be found in Appendix C.1. After processing all $K$ blocks, the final compressed approximation of $W$ is

$$W \approx \sum_{k=1}^{K} \widehat{P}^{(k)} \widehat{Q}^{(k)}. \tag{9}$$

## 4.2 Intra-Block Quantization Compensation

Even with block-wise decomposition, quantizing multiple components within a block may lead to accumulated errors. When the early components in a block are quantized, their distortion perturbs the effective subspace seen by the remaining unquantized components, degrading the overall reconstruction quality.

To address this, we introduce an *intra-block quantization compensation* strategy that explicitly corrects for these distortions. The key idea is to adjust the coefficients of the unquantized components within the same block in response to the quantization errors of earlier components. This is related to classical subspace projection methods (Golub and Van Loan, 2013; Saad, 2003), where updates are projected onto the subspace spanned by remaining basis vectors. By redistributing the quantization-induced error into the subspace of unquantized directions, we ensure that subsequent components adapt to the modified residual, reducing error accumulation and improving overall reconstruction fidelity. The technical details can be found in supplemental materials. We provide the quantization algorithm including block-wise structure and intra-block compensation in Algorithm 1.

This two-stage compression scheme offers both theoretical alignment with the optimal low-rank structure and practical robustness under quantization. The block-wise design improves error isolation and quantization efficiency, while intra-block compensation allows for precise error absorption. Together, they support high-accuracy approximation under low bitwidth constraints.

## 5 Experiments

In this section, we comprehensively evaluate the proposed LoRDQ framework on various large language models, including LLaMA-2 (Touvron et al., 2023), LLaMA-3 (Dubey et al., 2024), and OPT families (Zhang et al., 2022), across multiple benchmarks. We aim to highlight both the strengths and limitations of LoRDQ, as well as provide deeper insights into its behavior through ablation studies.

Our experimental setup closely follows the post-training quantization pipeline used in prior works such as GPTQ, AWQ. All experiments were conducted on a single NVIDIA A100 GPU with 80GB memory. We use HuggingFace implementations of the evaluated models. For calibration, we randomly sampled 128 segments of 2048 tokens each from the C4 dataset (Raffel et al., 2020). We focus on weight-only quantization, as this component dominates storage and transmission cost in large models. We report perplexity on WikiText2 (Merity et al., 2016), PTB (Marcus et al., 1994), and C4, as well as zero-shot accuracy on StoryCloze (Mostafazadeh et al., 2016), PIQA (Tata and

---

**Algorithm 1** Block-Wise Quantization with Compensation

---

**Require:** Weight matrix $W$, Cholesky factor $Y$, total rank $r$, block count $K$, quantizer `Quantize()`

**Ensure:** Quantized low-rank factors $\hat{P}, \hat{Q}$

1: Compute $W_Y = WY$, set $r_b \leftarrow r/K$, $R_Y \leftarrow W_Y$
2: **for** block $k = 1$ to $K$ **do**
3:    $[U, \Sigma, V] \leftarrow$ Top-$r_b$ SVD of $R_Y$
4:    $P^{(k)} = U\Sigma, Q^{(k)} = V^\top Y^{-1}$
5:    **for** $i = 1$ to $r_b$ **do**
6:       $\hat{p}_i = $ `Quantize`$(p_i), \hat{q}_i = $ `Quantize`$(q_i)$
7:       $\delta b = Y^\top(\hat{q}_i - q_i)$
8:       **for** $j = i + 1$ to $r_b$ **do**
9:          $\alpha = (Y^\top q_j)^\top \delta b$
10:         $p_j \leftarrow p_j - \alpha \cdot \hat{p}_i$
11:       **end for**
12:       Store $\hat{p}_i, \hat{q}_i$
13:    **end for**
14:    Update residual: $R_Y \leftarrow R_Y - \hat{P}^{(k)}\hat{Q}^{(k)}Y$
15: **end for**
16: **return** Concatenated $\hat{P}, \hat{Q}$

---

Table 1: Comparison of LoRDQ and GPTQ on various Llama2, Llama3, and OPT models and datasets with **2-bit quantization**. Perplexity ($\downarrow$) and accuracy ($\uparrow$) metrics are reported.

| Method | Model | Perplexity ($\downarrow$) | | | Accuracy ($\uparrow$) | | | | |
|---|---|---|---|---|---|---|---|---|---|
| | | C4 | WikiText2 | PTB | StoryCloze | PIQA | ARC-E | ARC-C | BoolQ |
| LoRDQ | Llama2-7B | 119.71 | 198.04 | 1.1e3 | 50.29 | 53.54 | 28.24 | 20.07 | 45.60 |
| GPTQ | Llama2-7B | 2.2e3 | 1.1e4 | - | 50.03 | 52.18 | 25.26 | 23.08 | 43.06 |
| AWQ | Llama2-7B | 2.2e5 | 1.7e5 | - | - | 52.39 | 24.75 | - | - |
| LoRDQ | Llama2-13B | 48.07 | 72.71 | 529.87 | 53.13 | 55.66 | 30.43 | 23.41 | 38.29 |
| GPTQ | Llama2-13B | 293.79 | 1.0e3 | 4.4e3 | 49.97 | 51.14 | 28.25 | 23.41 | 39.36 |
| AWQ | Llama2-13B | 1.2e5 | 9.5e4 | - | - | 53.26 | 23.04 | - | - |
| LoRDQ | Llama3-8B | 300.43 | 1.0e4 | 2.1e4 | 50.35 | 53.32 | 27.48 | 20.07 | 58.29 |
| GPTQ | Llama3-8B | 2.7e5 | 1.0e6 | 1.6e6 | 48.16 | 51.03 | 26.67 | 20.07 | 46.12 |
| AWQ | Llama3-8B | 8.1e5 | 8.2e5 | 9.0e5 | - | 55.2 | 25.2 | 21.3 | - |
| LoRDQ | OPT-13b | 28.27 | 41.92 | 38.01 | 65.31 | 65.56 | 48.42 | 26.76 | 38.10 |
| GPTQ | OPT-13b | 135.48 | 372.68 | 344.44 | - | 66.05 | 42.47 | - | - |

Patel, 2003), ARC-Easy (Boratko et al., 2018), ARC-C (Boratko et al., 2018), and BoolQ (Clark et al., 2019). For fair comparison across all methods, we do not use group-wise scaling factors but instead adopt a single scaling factor for the entire row vector of each weight matrix.

To ensure a fair comparison with existing PTQ approaches, we match the *average bits per weight (bpp)* across methods using the *stored quantized factors only*. Baselines (e.g., GPTQ, AWQ) quantize the full matrix at a uniform or binary bitwidth; in contrast, LoRDQ stores only low-rank factors at higher precision. Concretely, for $W \in \mathbb{R}^{M \times N}$ represented as $W \approx PQ$ with $P \in \mathbb{R}^{M \times r}$ and $Q \in \mathbb{R}^{r \times N}$, the resulting average bits/weight is $\text{bpp} = \frac{b_P\,Mr + b_Q\,rN}{MN}$. Unless otherwise stated, we use $b_P = b_Q = 4$ and choose $r$ *per layer* so that the model-level bpp matches a 2-bit/3-bit baseline when the numerator and denominator are summed over all layers. Under this setting on LLaMA-2-7B, we retain 31.6% of basis components on average across projection/MLP matrices, yielding an overall bpp of $\approx 2.00$ based on the stored factors $P, Q$ alone.

## 5.1 EFFICIENCY OF THE PROPOSED LOW-RANK DECOMPOSITION

We evaluate how accurately the decomposition preserves layer outputs as a function of rank. For a given rank $r$, we report the *average relative output error* across layers,

$$\mathrm{RelErr}(r) \;=\; \frac{\|(W - \widehat{W}_r)X\|_F^2}{\|WX\|_F^2},$$

and illustrate two representative layers by comparing two decompositions: (i) our *activation-aware* low-rank factors (LoRDQ), (ii) the classical SVD of $W$ (activation-agnostic). This comparison isolates the benefit of incorporating layer input activation. As shown in Figure 2, the *activation-aware* decomposition (LoRDQ) reduces error sharply with rank. For instance, the relative output error decreases to $50\%$ with $\sim 0.5\%$ relative rank, and reduce sharply from roughly $50\% \to 25\%$ by $\sim 10\%$ relative rank, indicating that it captures the dominant output-energy directions early. In side-by-side comparisons, LoRDQ consistently reach $10\%$ less output errors below the classical SVD of $W$ (activation-agnostic) at the same rank, demonstrating the efficiency of our method in extracting key components.

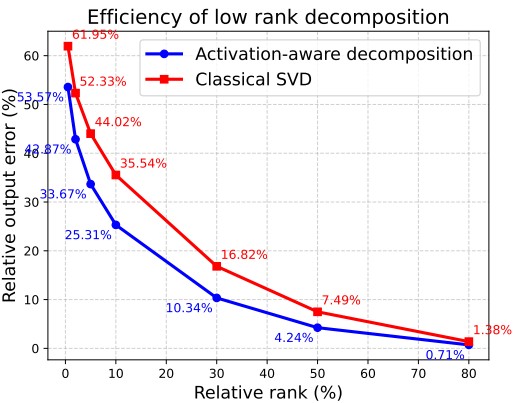

Figure 2: Efficiency of proposed decomposition vs. classical SVD

## 5.2 PERFORMANCE COMPARISON

Table 1 reports the performance of LoRDQ against strong baselines, including GPTQ and AWQ, under the 2-bit weight-only quantization regime. More results with OPT models can be found in Appendix D.1. Across all models and benchmarks, LoRDQ consistently outperforms GPTQ and AWQ, demonstrating the effectiveness of the proposed low-rank decomposition combined with block-wise greedy quantization and intra-block compensation. To interpret these observed gains, it has been checked by extensive simulations that the spectrum of output Gram matrix $WXX^\top W^\top$ is highly relevant to the efficiency of LoRDQ: concentrated spectra (few dominant components) in $WXX^\top W^\top$ imply higher compressibility and favor LoRDQ's precision allocation, whereas flatter spectra limit gains (see Appendix D.2 for more details).

When the average number of bits increase to 3, LoRDQ can underperform GPTQ/AWQ because it compresses two factors $(P, Q)$, introducing extra quantization noise compared to single-matrix methods when the bit budget is not tight (see Appendix D.3 for details). To verify the efficiency of our approach in 3-bit regime, we also consider another practical scenario inspired by transformation-based quantization schemes, where $Q$ is treated as a *quantization-free transformation matrix* stored in full precision. This configuration is relevant in cases where $Q$ serves as a learned transformation that can be precomputed without incurring significant storage overhead. By eliminating quantization noise in $Q$, the entire bit budget can be allocated to compressing $P$, thereby maximizing the representational capacity of the quantized factors. Table 2 shows the performance of LoRDQ under this *quantization-free* $Q$ configuration, alongside state-of-the-art transformation-based methods such as QuIP and OmniQuant. In this setting, LoRDQ achieves comparable or slightly better results than these approaches, demonstrating its competitiveness when applied as a transformation-based quantization framework.

Table 2: Comparison of LoRDQ and transformation-based methods on various models and datasets with **3-bit quantization**. The transformation matrix $Q$ is assumed to be quantization-free, thus we put all resources into quantizing $P$.

| Method | Model | Perplexity ($\downarrow$) | | | Accuracy ($\uparrow$) | | | | |
|---|---|---|---|---|---|---|---|---|---|
| | | C4 | WikiText2 | PTB | SC | PIQA | ARC-E | ARC-C | BoolQ |
| LoRDQ | Llama2-7B | 7.71 | 6.15 | 27.81 | 75.63 | 76.77 | 72.77 | 38.13 | 37.43 |
| QuIP | Llama2-7B | 20.44 | 18.66 | - | - | 65.45 | 56.57 | 25.68 | - |
| OmniQ | Llama2-7B | 8.62 | 6.62 | - | - | 74.65 | 71.00 | 38.14 | - |
| LoRDQ | Llama2-13B | 7.26 | 5.65 | 35.57 | 77.55 | 78.02 | 73.95 | 39.13 | 38.50 |
| QuIP | Llama2-13B | 7.16 | 5.61 | - | - | 77.31 | 75.38 | 42.66 | - |
| OmniQ | Llama2-13B | 7.39 | 5.58 | - | - | 77.97 | 76.60 | 43.34 | - |
| LoRDQ | Llama3-8B | 7.85 | 6.52 | 10.73 | 76.64 | 78.24 | 75.29 | 41.47 | 36.51 |
| QuIP | Llama3-8B | 11.70 | 8.48 | - | - | 75.79 | 72.01 | 39.68 | - |
| OmniQ | Llama3-8B | 20.36 | 14.70 | - | - | 68.12 | 59.68 | 28.16 | - |

Table 3: Comparison of SVD and SVD+Compensation methods with different block counts under **2-bit quantization**. Perplexity ($\downarrow$) and accuracy ($\uparrow$) metrics are reported.

| Block count | Method | Perplexity ($\downarrow$) | | | Accuracy ($\uparrow$) | | | | |
|---|---|---|---|---|---|---|---|---|---|
| | | C4 | WikiText2 | PTB | StoryCloze | PIQA | ARC-E | ARC-C | BoolQ |
| 1 | Block-wise decomposition | 5240.53 | 6221.77 | 8458.17 | 47.84 | 53.48 | 27.89 | 20.07 | 37.83 |
| | Block-wise + Intra-block | 9887.89 | 14610.08 | 18631.13 | 47.78 | 53.70 | 28.42 | 21.40 | 39.72 |
| 2 | Block-wise decomposition | 85.57 | 143.68 | 748.19 | 52.27 | 54.03 | 32.28 | 22.07 | 37.83 |
| | Block-wise + Intra-block | 48.07 | 72.71 | 529.87 | 53.13 | 55.66 | 30.43 | 23.41 | 38.29 |
| 3 | Block-wise decomposition | 46.53 | 77.87 | 447.09 | 53.93 | 55.60 | 32.46 | 23.75 | 39.42 |
| | Block-wise + Intra-block | 45.08 | 74.19 | 410.50 | 54.78 | 56.31 | 32.98 | 22.41 | 37.89 |
| 4 | Block-wise decomposition | 45.40 | 77.47 | 442.44 | 55.05 | 56.47 | 35.79 | 23.75 | 38.10 |
| | Block-wise + Intra-block | 47.24 | 79.19 | 456.78 | 55.42 | 57.02 | 33.33 | 25.08 | 38.20 |

## 5.3 ABLATION STUDY

We analyze the effects of the proposed block-wise decomposition and intra-block compensation (Table 3). Increasing the number of blocks significantly improves performance, as it allows the residual structure to be progressively refined, leading to lower perplexity and higher accuracy across tasks. In contrast, the intra-block compensation mechanism provides only marginal gains, with its influence being less substantial compared to the impact of block-based decomposition. These results highlight that block-wise decomposition is the primary driver of performance improvements in LoRDQ, while compensation plays a secondary role in improving the reconstruction quality under aggressive quantization. To better understand the trade-off between compressing $P$ and $Q$, we vary their bitwidths $(N_P, N_Q)$ under a fixed overall budget to conduct another ablation study in Appendix E.

## 6 CONCLUSION

We introduced LoRDQ, an activation-aware low-rank decomposition and quantization framework for post-training compression of large language models. By deriving a closed-form activation-aware low-rank decomposition and integrating block-wise greedy decomposition with intra-block quantization compensation, LoRDQ achieves strong compression–accuracy trade-offs, particularly under ultra-low bitwidth constraints. Extensive experiments on LLaMA-2, LLaMA-3, and OPT models demonstrate that LoRDQ delivers substantial improvements over state-of-the-art PTQ methods in ultra low-bit regimes, while remaining competitive with transformation-based schemes in higher-bit settings by ignoring the overhead of transformation. Our analysis connects the observed gains to the spectral properties of the Gram matrix of layer outputs, providing a theoretical interpretation of why low-rank structures enable efficient quantization. These insights motivate a potential hybrid integration with existing quantization: deploy LoRDQ on layers with concentrated spectra to lift performance, and defer to standard schemes on flat-spectrum layers to avoid degradation. These results highlight LoRDQ as an effective and interpretable framework for scaling large models to resource-constrained environments.

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

APPENDIX

## A  PROOF OF PROPOSITION 1

Consider the optimization problem

$$\min_{P\in\mathbb{R}^{M\times r}, Q\in\mathbb{R}^{r\times N}} \|(W - PQ)X\|_F^2, \tag{10}$$

where $r < \min(M, N)$ specifies the target rank of the approximation. Expanding the Frobenius norm in trace form gives

$$\|(W - PQ)X\|_F^2 = \mathrm{Tr}\big((W - PQ)XX^\top(W - PQ)^\top\big). \tag{11}$$

Let $XX^\top = YY^\top$ be any square-root factorization of the input covariance with $Y \in \mathbb{R}^{N\times N}$ invertible. Substituting this into the objective yields

$$\|(W - PQ)X\|_F^2 = \|(W - PQ)Y\|_F^2. \tag{12}$$

Defining $W_Y := WY \in \mathbb{R}^{M\times N}$ and $Q_Y := QY \in \mathbb{R}^{r\times N}$, the problem becomes

$$\min_{P\in\mathbb{R}^{M\times r}, Q_Y\in\mathbb{R}^{r\times N}} \|W_Y - PQ_Y\|_F^2, \tag{13}$$

which is a standard low-rank approximation problem for $W_Y$.

By the Eckart–Young–Mirsky theorem, the minimizer of this problem is given by the rank-$r$ truncated SVD of $W_Y$:

$$W_Y \approx U_r\Sigma_r V_r^\top, \tag{14}$$

where $U_r \in \mathbb{R}^{M\times r}$, $\Sigma_r \in \mathbb{R}^{r\times r}$, and $V_r \in \mathbb{R}^{N\times r}$ are the top-$r$ singular vectors and singular values of $W_Y$. Thus, the optimal factors are

$$P^* = U_r\Sigma_r, \quad Q_Y^* = V_r^\top. \tag{15}$$

Recovering $Q$ from $Q_Y = QY$ gives

$$Q^* = V_r^\top Y^{-1}. \tag{16}$$

Finally, note that any other valid square-root factorization of $XX^\top$ only changes $V_r$ by an orthogonal rotation, which leaves the Frobenius norm of the approximation invariant. Therefore, $(P^*, Q^*)$ also solves the original problem with $X$, achieving its global minimum.

## B  PROOF OF COROLLARY 1

Consider the broader low-rank approximation problem

$$\min_{A\in\mathbb{R}^{M\times r}, B\in\mathbb{R}^{r\times K}} \|WX - AB\|_F^2. \tag{17}$$

By the Eckart–Young–Mirsky theorem, its optimal solution is the rank-$r$ truncated SVD of $WX$, which we denote as $WX \approx P_X Q_X$. Thus, the minimizers of this problem are $A^* = P_X$ and $B^* = Q_X$, where $P_X \in \mathbb{R}^{M\times r}$ and $Q_X \in \mathbb{R}^{r\times K}$ are the top-$r$ singular factors of $WX$.

From Proposition 1, for any square-root factorization $XX^\top = YY^\top$, the optimal solution to

$$\min_{P\in\mathbb{R}^{M\times r}, Q\in\mathbb{R}^{r\times N}} \|(W - PQ)X\|_F^2 \tag{18}$$

is given by $P^* = U_r\Sigma_r$ and $Q^* = V_r^\top Y^{-1}$, where $U_r\Sigma_r V_r^\top$ is the truncated SVD of $W_Y := WY$.

To connect these two problems, define $X' := Y^{-1}X$. Then $WX = (WY)X' = W_Y X'$. Note that

$$X'X'^\top = Y^{-1}XX^\top Y^{-T} = Y^{-1}YY^\top Y^{-T} = I, \tag{19}$$

so $X'$ has orthonormal rows. This orthonormality implies that post-multiplying $W_Y$ by $X'$ does not change the dominant left singular subspace: the rank-$r$ truncated SVD of $W_Y$ followed by multiplication with $X'$ yields exactly the rank-$r$ truncated SVD of $WX$. Therefore,

$$P^*Q^*X = U_r\Sigma_r V_r^\top Y^{-1}X = P_X Q_X. \tag{20}$$

This shows that the pair $(P^*, Q^*)$ from Proposition 1 produces the same rank-$r$ reconstruction of $WX$ as directly solving the broader problem, thereby achieving the minimum.

## C  DETAILS OF COMPRESSION TECHNIQUES

### C.1  BLOCK-WISE DECOMPOSITION

To produce these quantized factors, we use a scalar uniform quantizer $\mathrm{Quantize}(\cdot)$ with $b$-bit precision. For a vector $z \in \mathbb{R}^d$, its quantized version is defined as

$$\widehat{z} = \mathrm{Quantize}(z) = \mathrm{clip}\Big(\mathrm{round}\Big(\frac{z-\alpha}{\Delta}\Big), 0, 2^b - 1\Big)\Delta + \alpha, \tag{21}$$

where $\Delta = \frac{\beta-\alpha}{2^b-1}$ is the quantization step size, and $\alpha, \beta$ are the per-vector minimum and maximum values (or learned bounds).

After obtaining the residual, we compute the rank-$r_b$ SVD of the residual:

$$\widetilde{R}_Y^{(k)} = U_k \Sigma_k V_k^\top, \tag{22}$$

where $U_k \in \mathbb{R}^{M \times r_b}$, $\Sigma_k \in \mathbb{R}^{r_b \times r_b}$, and $V_k \in \mathbb{R}^{N \times r_b}$. The unquantized block factors in the projected space are

$$P_Y^{(k)} := U_k \Sigma_k, \quad Q_Y^{(k)} := V_k^\top. \tag{23}$$

These are mapped back to the original parameter space:

$$P^{(k)} := P_Y^{(k)}, \quad Q^{(k)} := Q_Y^{(k)} Y^{-1}, \tag{24}$$

and then quantized:

$$\widehat{P}^{(k)} = \mathrm{Quantize}(P^{(k)}), \quad \widehat{Q}^{(k)} = \mathrm{Quantize}(Q^{(k)}). \tag{25}$$

After processing all $K$ blocks, the final compressed approximation of $W$ is

$$W \approx \sum_{k=1}^{K} \widehat{P}^{(k)} \widehat{Q}^{(k)}. \tag{26}$$

### C.2  INTRA-BLOCK COMPENSATION

Formally, for block $k$, the rank-$r_b$ approximation of $R_Y^{(k)}$ is

$$\widetilde{R}_Y^{(k)} = \sum_{i=1}^{r_b} p_i^{(k)} (q_i^{(k)})^\top Y,$$

where $p_i^{(k)}$ is the $i$-th column of $P^{(k)}$ and $q_i^{(k)}$ the $i$-th row of $Q^{(k)}$. Define the orthogonalized right factors:

$$b_i^{(k)} := Y^\top q_i^{(k)}.$$

The block can then be written as

$$\widetilde{R}_Y^{(k)} = \sum_{i=1}^{r_b} p_i^{(k)} (b_i^{(k)})^\top.$$

Quantize each component sequentially:

$$\hat{p}_i^{(k)} = \mathrm{Quantize}(p_i^{(k)}), \quad \hat{q}_i^{(k)} = \mathrm{Quantize}(q_i^{(k)}),$$

and define the error in the orthogonal basis:

$$\delta b_i^{(k)} := \hat{b}_i^{(k)} - b_i^{(k)}, \quad \hat{b}_i^{(k)} := Y^\top \hat{q}_i^{(k)}.$$

Project this error onto the subspace spanned by the unquantized components:

$$\alpha_{ij}^{(k)} := (b_j^{(k)})^\top \delta b_i^{(k)}, \quad \forall i < j \le r_b,$$

and update subsequent left factors:

$$p_j^{(k)} \leftarrow p_j^{(k)} - \alpha_{ij}^{(k)} \hat{p}_i^{(k)}.$$

This redistribution of quantization-induced distortion reduces intra-block error accumulation, significantly improving reconstruction fidelity under low-bitwidth constraints.

Table 4: Comparison of LoRDQ, GPTQ and AWQ on various OPT models and datasets with **2-bit quantization**. Perplexity (↓) and accuracy (↑) metrics are reported.

| Method | Model | Perplexity (↓) | | | Accuracy (↑) | | | | |
|--------|-------|------|-----------|------|------|------|-------|-------|-------|
| | | C4 | WikiText2 | PTB | SC | PIQA | ARC-E | ARC-C | BoolQ |
| LoRDQ | OPT-125M | 335.18 | 840.40 | 864.13 | 51.52 | 54.08 | 28.62 | 19.06 | 39.24 |
| GPTQ | OPT-125M | 2161.69 | 4444.83 | 3072.12 | 48.58 | 54.41 | 27.89 | 20.74 | 39.08 |
| LoRDQ | OPT-350M | 538.87 | 1.7e3 | 1.3e3 | 50.72 | 53.81 | 28.20 | 21.40 | 37.83 |
| GPTQ | OPT-350M | 5548.46 | 15608.65 | 10147.67 | 48.58 | 52.88 | 29.65 | 22.74 | 38.07 |
| LoRDQ | OPT-1.3B | 232.86 | 447.84 | 541.70 | 50.03 | 54.73 | 30.47 | 19.73 | 42.94 |
| GPTQ | OPT-1.3B | 3373.44 | 8171.65 | 5745.94 | 48.53 | 53.26 | 27.54 | 21.40 | 46.73 |
| AWQ | OPT-1.3B | 6.4e3 | 9.5e3 | 5.9e3 | - | 51.63 | 24.83 | 20.05 | 37.82 |
| LoRDQ | OPT-2.7B | 123.01 | 234.27 | 244.73 | 53.93 | 55.17 | 34.05 | 19.73 | 41.71 |
| GPTQ | OPT-2.7B | 3898.29 | 9346.39 | 5904.72 | 47.57 | 53.37 | 28.07 | 18.73 | 38.53 |
| AWQ | OPT-2.7B | 1.2e4 | 2.3e4 | 9.0e3 | - | 53.15 | 25.04 | 21.67 | 40.09 |
| LoRDQ | OPT-6.7B | 61.97 | 90.99 | 116.16 | 57.46 | 59.25 | 38.93 | 21.40 | 39.27 |
| GPTQ | OPT-6.7B | 489.35 | 3270.47 | 2605.91 | 51.15 | 54.73 | 32.98 | 21.40 | 38.87 |

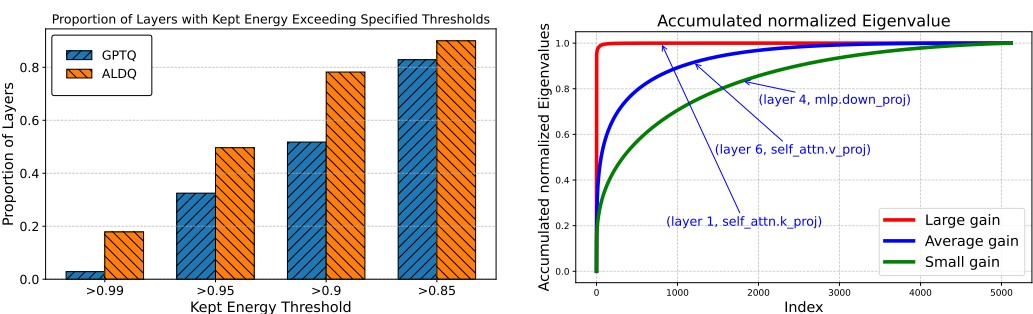

(a) Retained energy ratio: LoRDQ vs. GPTQ (LLaMA2-13B).

(b) Spectral profile: cumulative energy vs. components.

Figure 3: Spectral comparisons for LLaMA2-13B. Layers with more concentrated spectra (higher energy in top components) tend to yield larger LoRDQ gains over GPTQ.

# D  ADDITIONAL EXPERIMENTS

## D.1  2-BIT PERFORMANCE COMPARISON WITH OPT MODELS

Table 4 reports the 2-bit quantization performance of LoRDQ compared to GPTQ and AWQ across OPT models of various sizes. LoRDQ consistently achieves substantially lower perplexity and higher or comparable accuracy across all benchmarks. This demonstrates the effeciency of our mehtod in different kind of models with different size.

## D.2  EXPLANATIONS OF THE OBSERVED GAINS

To better interpret these gains, we quantify how much of the layer output is preserved by different quantization schemes. Specifically, we define the *retained energy ratio* of a method $i \in \{\text{LoRDQ}, \text{GPTQ}\}$ as

$$\gamma^{(i)} = \left(1 - \frac{\|(W - \widehat{W}^{(i)})X\|_F^2}{\|WX\|_F^2}\right) \times 100\%, \tag{27}$$

which measures the fraction of the output energy retained after quantization. A higher $\gamma^{(i)}$ indicates a more efficient compression, leading to lower reconstruction loss in the layer outputs. Figure 2 compares this retained energy ratio for LoRDQ and GPTQ on the LLaMA2-13B model, showing that LoRDQ consistently achieves higher values across layers, reflecting its ability to preserve more of the informative structure of the weight matrices. This observation aligns with the superior perplexity and accuracy results in Table 1.

To further investigate why LoRDQ achieves these gains, we analyze the spectrum of $WXX^\top W^\top$, which directly determines the retained output energy. As shown in Figure 3, layers with concentrated spectra (where a

few dominant components capture most of the energy) are more suited for low-rank decomposition, whereas flatter spectra indicate limited compressibility. This analysis reveals that LoRDQ performs better in in layers with concentrated spectra by effectively allocating precision to the dominant components, explaining its advantage over GPTQ in preserving key information under the same bit budget.

## D.3 SOME RESULTS AT 3-BIT REGIME

Table 5: Comparison of LoRDQ and GPTQ on various Llama2, Llama3, and OPT models and datasets with **3-bit quantization**. The second half, the transformation matrix $Q$ is assumed to be quantization-free, thus we put all resources into quantizing $P$. Perplexity ($\downarrow$) and accuracy ($\uparrow$) metrics are reported.

| | | Perplexity ($\downarrow$) | | | Accuracy ($\uparrow$) | | | | |
|---|---|---|---|---|---|---|---|---|---|
| **Method** | **Model** | C4 | WikiText2 | PTB | SC | PIQA | ARC-E | ARC-C | BoolQ |
| LoRDQ | Llama2-7B | 48.44 | 72.76 | 443.93 | 54.52 | 57.78 | 33.84 | 25.08 | 41.22 |
| GPTQ | Llama2-7B | 10.39 | 9.50 | 7.3e3 | 71.51 | 70.78 | 60.18 | 31.44 | 38.26 |
| AWQ | Llama2-7B | 23.85 | 24.00 | - | - | 65.02 | 52.78 | - | - |
| LoRDQ | Llama2-13B | 19.75 | 22.91 | 241.94 | 62.96 | 64.31 | 47.94 | 25.42 | 38.04 |
| GPTQ | Llama2-13B | 8.24 | 6.78 | 49.92 | 74.88 | 73.83 | 68.07 | 36.12 | 41.13 |
| AWQ | Llama2-13B | 13.07 | 10.45 | - | - | 70.13 | 66.79 | - | - |
| LoRDQ | Llama3-8B | 52.66 | 210.80 | 323.15 | 52.00 | 55.55 | 30.35 | 21.40 | 37.92 |
| GPTQ | Llama3-8B | 29.87 | 81.82 | 70.88 | 53.50 | 58.92 | 36.49 | 22.07 | 38.65 |
| AWQ | Llama3-8B | 16.80 | 12.80 | 24.00 | - | 71.90 | 66.70 | 35.10 | - |

As shown in Table 5, LoRDQ performs worse than GPTQ and AWQ in the 3-bit quantization setting, marking a clear contrast with its relative advantage in the ultra-low 2-bit regime. This performance gap arises from LoRDQ's design: unlike conventional methods that quantize a single full-rank weight matrix, LoRDQ compresses two low-rank factors, $P$ and $Q$. While this approach offers substantial benefits under very low bit budgets, it also introduces additional quantization noise, which can offset these gains when the available bitwidth is relatively high (e.g., 3 bits). In such cases, the double-matrix compression limits overall performance.

# E ABLATION STUDY OF BITS ALLOCATION

To better understand the trade-off between compressing $P$ and $Q$, we vary their bitwidths $(N_P, N_Q)$ under a fixed overall budget (Table 6). The results show that the $(4, 4)$ configuration provides the best overall trade-off, achieving strong performance across tasks. Allocating fewer bits per factor (e.g., $(3, 3)$ or $(3, 4)$) allows a higher retained rank but results in substantially lower quality, whereas higher precision with lower rank (e.g., $(5, 5)$) preserves quality but at the cost of rank diversity. These results illustrate the balance between retained rank and per-factor precision, with $(4, 4)$ offering the most favorable compromise in this setting.

Table 6: Effect of different $(N_P, N_Q)$ settings on LLaMA-2-13B with low-rank SVD decomposition under a fixed average 2-bit-per-weight budget. The column "Retained rank ($r/N$)" indicates the proportion of singular components preserved relative to the full rank $N$. This setup ensures that the total number of bits used for compressing the network remains constant, balancing between retaining fewer components at higher precision or more components at lower precision.

| $(N\_P, N\_Q)$ | Retained rank ($r/N$) | wikitext | ptb | c4 | StoryCloze | PIQA | ARC | ARC-C | BoolQ |
|---|---|---|---|---|---|---|---|---|---|
| (5,5) | 25.3% | 882.23 | 2471.30 | 543.37 | 50.72 | 51.85 | 27.89 | 21.07 | 37.83 |
| (4,4) | 31.6% | 72.71 | 529.87 | 48.07 | 53.13 | 55.66 | 30.43 | 23.41 | 38.29 |
| (3,4) | 36.7% | 1117.26 | 3071.91 | 866.69 | 49.76 | 52.94 | 27.61 | 19.73 | 37.83 |
| (4,3) | 35.4% | 389.36 | 1768.70 | 239.82 | 51.31 | 53.21 | 27.90 | 19.73 | 37.83 |
| (3,3) | 42.2% | 9274.93 | 7959.67 | 7174.23 | 48.32 | 52.29 | 27.54 | 20.40 | 37.83 |

