# OpenReview forum: "LoRDQ: activation-aware Low-Rank Decomposition and Quantization for Large Language Model Compression"
_ICLR.cc/2026/Conference — ICLR 2026 Conference Withdrawn Submission_

### Official Review · Reviewer_6H1S · 2025-10-29

**Soundness:** 2
**Presentation:** 2
**Contribution:** 2
**Rating:** 2
**Confidence:** 4

**Summary:**

The authors formulate an activation-aware low-rank decomposition that minimizes layer output error and derive a closed-form solution for any valid square-root factorization of the Gram matrix of layer inputs, which leads to proposing LoRDQ. LoRDQ involves two complementary techniques to mitigate the loss from quantizing low-rank factors, including a block-wise greedy decomposition and an intra-block compensation step.

**Strengths:**

1. The authors combines a block-wise greedy decomposition with an intra-block quantization compensation strategy to mitigate quantization-induced performance degradation under aggressive bit-width constraints.
2. They conduct extensive experiments across multiple LLMs under 2-bit or 3-bit quantization schemes.

**Weaknesses:**

1. In Table 1, the baselines (only GPTQ and AWQ) are too limited to justify the effectiveness of the proposed method under 2-bit quantization. Given that QuIP is effective for 2-bit quantization, it is necessary to compare the proposed method with QuIP.
2. In Table 1, LoRDQ shows better performance than GPTQ and AWQ, but the ppl of LoRDQ is still too high, which seems not practical at all.
3. Most LLMs used in Table 1-3 are old-fashioned. Considering that low-bit quantization favors under-trained LLMs [1], it is required to see whether the proposed method is also effective for recent models such as Llama 3.2, Qwen2.5, or Qwen3.
4. Either PPL or CSR accuracy is reported. So, it is hard to determine whether LoRDQ performs well or not. It would be more beneficial if the authors also explore more challenging tasks (e.g., MMLU) and generation tasks (e.g., IFEval, GSM8K) with recent LLMs mentioned in Weakness 3.
5. As the bit-width to preserve the performance of the original FP model is also important, can the authors provide the experimental results under 4-bit quantization?
6. In Table 3, it is hard to understand why intra-block quantization compensation is sometimes helpful and sometimes not. Can the authors provide more detailed analysis about this?

[1] Low-Bit Quantization Favors Undertrained LLMs, ACL 2025.

**Questions:**

In Table 3, the performance goes up as the number of blocks increases. Then, when does the performance saturate?

---

### Official Review · Reviewer_d9MU · 2025-10-30

**Soundness:** 2
**Presentation:** 2
**Contribution:** 2
**Rating:** 2
**Confidence:** 4

**Summary:**

The paper propose a way to compress weights for large language models by performing joint low rank decomposition and quantization. The paper is well written. But it does not fairly characterize the contributions in comparison to baselines. For example, the activation aware low rank decomposition is similar to ASVD, the residual compression is similar in to ResSVD. What if I create a new baseline which applies GPTQ/AWQ over ASVD or ResSVD? Is that application trivial? How does LoRDQ compare with such a baseline. Further, the evaluations in this paper are a little weak and does not consider complex tasks or hardware numbers.

**Strengths:**

LordQ outperforms GPTQ and AWQ baselines at 2-bit average per weight and Omni-quant, QUIP baselines at 3-bit average bits per weight.

**Weaknesses:**

1. The source code is not provided.
2. The baselines considered are relatively old. The paper lacks comparison with more recent baselines.
3. The paper lacks larger open source model evaluations (eg. Llama-3-70B).
4. The paper lacks recent evaluation benchmarks (eg. code completion, long context reasoning, mathematical understanding, etc.)
5. The paper lacks hardware evaluations/speedup with the technique.
6. The paper does not report time/computational resources required to compress LLMs using LordQ.

**Questions:**

1. How beneficial is using Cholesky decomposition ($XX^T = LL^T$) ? Paper mentions that computation complexity is reduced from $O(N^3)$ to $O(N^3/3)$. How big of an issue is the computation complexity? Since it is a post training compression approach, this is a one time cost.
2. What if I apply GPTQ/AWQ quantization over ASVD or ResSVD low rank decomposition, how would LordQ compare with such a baseline.
3. How is the process of inter/intra block compression different from what GPTQ does for quantization.
4. What is the impact of calibration data on the compressed model?

---

### Official Review · Reviewer_JYD3 · 2025-10-30

**Soundness:** 2
**Presentation:** 3
**Contribution:** 2
**Rating:** 2
**Confidence:** 3

**Summary:**

This paper proposes LoRDQ, a post-training compression framework for large language models that combines activation-aware low-rank decomposition with efficient quantization. This paper proposes LoRDQ, a post-training compression framework for large language models that combines activation-aware low-rank decomposition with efficient quantization.

**Strengths:**

This paper demonstrates high originality by creatively combining low-rank decomposition and quantization into a unified, activation-aware framework, a novel synthesis beyond existing isolated approaches.
The quality is robust, featuring rigorous theoretical proofs (e.g., optimality of the Cholesky-based solution) and extensive empirical validation across major LLMs.
The clarity is excellent, with a logically structured narrative that effectively explains the complex interplay between decomposition and quantization.
Its significance is substantial, delivering state-of-the-art performance in the challenging ultra-low (2-bit) precision regime, which is critical for the practical deployment of LLMs on resource-constrained devices.

**Weaknesses:**

This paper has several notable weaknesses.

First, its performance drops significantly at 3-bit, where it is outperformed by GPTQ/AWQ. The authors attribute this to "extra quantization noise" from compressing two matrices, but this undermines the method's general applicability and suggests it is primarily a solution for extreme, sub-2-bit scenarios.

Second, a major omission is the lack of hardware efficiency analysis. The computational overhead of the Cholesky decomposition, block-wise SVD, and intra-block compensation is not discussed, leaving its practical deployment cost unknown.

Finally, the comparison to hybrid methods like QLoRA or QuIP# is insufficient; benchmarking against these would better situate its novelty in the rapidly evolving field of LLM compression.

**Questions:**

(1) The paper emphasizes the computational efficiency of the Cholesky decomposition over SVD but does not report any latency, throughput, or memory footprint measurements during inference. Could you provide a comparative analysis of the end-to-end inference speed and memory usage of a model compressed with LoRDQ versus strong baselines like GPTQ or AWQ?

(2) The performance drop in the 3-bit regime is a significant limitation. You note this is due to quantizing two matrices, but this suggests LoRDQ is only optimal for ultra-low-bit scenarios. Have you considered a hybrid or adaptive strategy where the method dynamically chooses between a) applying LoRDQ or b) falling back to a standard PTQ method like GPTQ on a per-layer basis, perhaps based on the spectral concentration metric you analyzed? Demonstrating such an adaptive system would greatly enhance the method's generality and practical utility.

---

### Official Review · Reviewer_jpvA · 2025-11-10

**Soundness:** 2
**Presentation:** 3
**Contribution:** 2
**Rating:** 2
**Confidence:** 4

**Summary:**

This paper presents LoRDQ, an activation-aware low-rank decomposition & quantization framework for post-training compression of large language models (LLMs). Unlike conventional post-training quantization (PTQ) methods that handle weights directly or assign precision heuristically, LoRDQ first derives a closed-form activation-aware low-rank decomposition based on the Cholesky factorization of the input covariance matrix, then quantizes the residuals again and again via a block-wise greedy decomposition and intra-block quantization compensation. Theoretically, the authors prove that their decomposition is equivalent to the optimal truncated SVD of the layer outputs without explicit whitening. Empirically, LoRDQ demonstrates consistent gains over GPTQ and AWQ in 2-bit quantization across LLaMA-2/3 and OPT models, achieving up to ~10× lower perplexity and higher zero-shot accuracy.

**Strengths:**

1.LoRDQ shows substantial performance advantages in ultra-low (2-bit) regimes, outperforming GPTQ/AWQ by large margins in perplexity and accuracy on multiple benchmarks and models.
2.Efficient and interpretable design: The Cholesky-based projection yields a closed-form and computationally efficient solution.

**Weaknesses:**

In general, the idea using covariance matrix to do SVD is not new, for example https://proceedings.neurips.cc/paper/2021/file/f56de5ef149cf0aedcc8f4797031e229-Paper.pdf
The author extend it to blockwise. However,  the study of block-wise is not clear. How K change in terms of error?

I think there is an important baseline missing in the experiments, which is the pure low rank,$W ≈\sum_{k=1}^KPQ$ where P,Q is low rank by svd, as the paper trying to sell the activation aware.

Others:
1. The paper rigidly fixes the calibration setup to 128 samples of 2048 tokens from the C4 dataset. However, since the covariance estimation directly affects the activation-aware decomposition, the choice of calibration data could significantly impact performance. I would like to see ablations over different dataset sources, sample counts, and sequence lengths, as well as authors’ recommendations on selecting appropriate covariance datasets for practical deployment.

2. Experiments focus on LLaMA and OPT families. Including more diverse architectures (e.g., Qwen-like, DeepSeek-like) or other quantization baselines (e.g., OmniQuant#) would better validate generality.
The rank r and bitwidth settings for P and Q appear to be selected empirically to match a 2-bit average without clear justification or adaptive strategy.

**Questions:**

1. LoRDQ’s decomposition explicitly depends on the input covariance How sensitive is the method to the choice of calibration dataset and token distribution? Can you do some ablation on it?

2.The paper focuses on 2- and 3-bit quantization, but the proposed method can be independently of the quantization. Better application needed to be found.

3. The results show that QuIP and OmniQuant outperform LoRDQ in ARC-E and ARC-C for LLaMA-3 models. Could the authors analyze the cause of this discrepancy—e.g., differences in dataset calibration, rank allocation, or architecture compatibility?

4. Does the two-factor representation (P and Q) increase matrix multiplication overhead during inference?
In intra-block compensation (Algorithm 1), how are the quantization errors redistributed—does this step significantly increase computational cost during compression?

5. Have the authors tested LoRDQ on reasoning-heavy or multilingual benchmarks (e.g., GSM8K, BBH, XWinograd), which might prevent inference speed up? Also I want to see more general benchmark results like MMLU, etc. Not only the 4 selected benchmark.

What’s ALDQ in Figure3.(a)
minor:
line 391 reach -> reaches
757 “better in in” -> in in
All 2bit -> 2-bits

---

### Note · Authors · 2025-11-25

I have read and agree with the venue's withdrawal policy on behalf of myself and my co-authors.